# Characterization of SARS-CoV-2 Escape Mutants to a Pair of Neutralizing Antibodies Targeting the RBD and the NTD

**DOI:** 10.3390/ijms23158177

**Published:** 2022-07-25

**Authors:** Antonia Sophia Peter, Eva Grüner, Eileen Socher, Kirsten Fraedrich, Elie Richel, Sandra Mueller-Schmucker, Arne Cordsmeier, Armin Ensser, Heinrich Sticht, Klaus Überla

**Affiliations:** 1Institute of Clinical and Molecular Virology, Universitätsklinikum Erlangen, Friedrich-Alexander University Erlangen-Nürnberg, 91054 Erlangen, Germany; antoniasophia.peter@uk-erlangen.de (A.S.P.); eva.gruener@med.uni-muenchen.de (E.G.); eileen.socher@fau.de (E.S.); kirsten.fraedrich@uk-erlangen.de (K.F.); elie.richel@uk-erlangen.de (E.R.); sandra.mueller-schmucker@uk-erlangen.de (S.M.-S.); arne.cordsmeier@uk-erlangen.de (A.C.); armin.ensser@fau.de (A.E.); 2Division of Bioinformatics, Institute of Biochemistry, Friedrich-Alexander University Erlangen-Nürnberg, 91054 Erlangen, Germany; heinrich.sticht@fau.de; 3Institute of Anatomy, Functional and Clinical Anatomy, Friedrich-Alexander University Erlangen-Nürnberg, 91054 Erlangen, Germany

**Keywords:** SARS-CoV-2, spike protein, NTD, RBD, variants of concern, SARS-CoV-2 escape mutations, neutralizing monoclonal antibodies

## Abstract

Mutations in the spike protein of SARS-CoV-2 can lead to evasion from neutralizing antibodies and affect the efficacy of passive and active immunization strategies. Immunization of mice harboring an entire set of human immunoglobulin variable region gene segments allowed to identify nine neutralizing monoclonal antibodies, which either belong to a cluster of clonally related RBD or NTD binding antibodies. To better understand the genetic barrier to emergence of SARS-CoV-2 variants resistant to these antibodies, escape mutants were selected in cell culture to one antibody from each cluster and a combination of the two antibodies. Three independently derived escape mutants to the RBD antibody harbored mutations in the RBD at the position T478 or S477. These mutations impaired the binding of the RBD antibodies to the spike protein and conferred resistance in a pseudotype neutralization assay. Although the binding of the NTD cluster antibodies were not affected by the RBD mutations, the RBD mutations also reduced the neutralization efficacy of the NTD cluster antibodies. The mutations found in the escape variants to the NTD antibody conferred resistance to the NTD, but not to the RBD cluster antibodies. A variant resistant to both antibodies was more difficult to select and only emerged after longer passages and higher inoculation volumes. VOC carrying the same mutations as the ones identified in the escape variants were also resistant to neutralization. This study further underlines the rapid emergence of escape mutants to neutralizing monoclonal antibodies in cell culture and indicates the need for thorough investigation of escape mutations to select the most potent combination of monoclonal antibodies for clinical use.

## 1. Introduction

Monoclonal neutralizing antibodies are a straightforward approach for the development of antiviral drugs. They were the first class of antivirals to receive emergency use approval against COVID-19 caused by SARS-CoV-2 [1]. This beta coronavirus, carrying a linear (+)ssRNA genome, is like all RNA viruses prone to errors during template copying [2]. Even though coronaviruses have a proof-reading function and therefore display a lower mutation rate than other RNA viruses, several new viral variants are and have been discovered since the start of the pandemic in December 2019 [3,4,5]. These new variants are classified by the world health organization (WHO) into different categories according to several criteria that amongst others consider their transmissibility, increase in virulence, or clinical presentation. The most alarming ones are categorized as variants of concern (VOC) [6]. The mutations do not only contribute to the immune evasion of the virus but also to the loss of effectivity of vaccines against virus variants, such as the Omicron VOC [7,8,9,10,11]. Consequently, it has also been shown that the neutralization of SARS-CoV-2 variants mediated by monoclonal antibodies can be severely impaired by mutations within the spike protein (S) [12,13,14,15,16].

Mutations that affect monoclonal antibody-mediated neutralization are often found within the receptor binding domain (RBD) and the N-terminal domain (NTD). The RBD is the interaction site of the spike protein with the cellular receptor human angiotensin converting enzyme 2 (hACE2). To date, several different mutations have been identified that alter the receptor–virus interaction. Here the T478K and S477N mutations, found for example in the Omicron variant Spike protein, are of great importance as they increase the stability and strength of the RBD–hACE2 interaction [17,18]. The RBD exposure itself, however, is most likely regulated by the NTD. Through its hypervariable loops, the NTD domain has been proposed to be involved in the control of S protein stability, membrane fusion potential, and RBD exposure [19,20]. Several different mutations have been identified in different VOC and virus variants that abrogate antibody binding of monoclonal antibodies interacting with the NTD. Here, for example, the Δ69–70 and ΔY144 mutations that are found amongst other NTD mutations in the Omicron and Alpha variants or the D215G, Δ242–244, and R246I NTD mutations identified in the Beta variant play an important role [21,22,23,24]. *De novo* mutations in SARS-CoV-2 do however not only arise in vivo but can also be observed in vitro [25]. It was for example described that the H245R mutation located within the NTD can be detected after eight passages on Vero-E6 cells [26]. Deletions and mutations in proximity of the furin cleavage site have also been observed frequently after repeated passaging of a virus on cells in vitro. These mutations are often linked to the adaptation of the virus to the furin levels expressed by the host cells [26]. Due to the highly dynamic mutational activity of the SARS-CoV-2 S protein, it is essential to fully understand the interaction of novel monoclonal, neutralizing antibodies with SARS-CoV-2 in order to be able to correctly predict the therapeutic potential.

We previously generated antibodies directed against SARS-CoV-2 S by repeated immunization of mice, carrying an entire set of human immunoglobulin variable region gene segments, with the S protein. After the generation of hybridoma, nine neutralizing antibodies belonging to two different clusters could be identified. The first cluster is comprised of three clonally related antibodies that interact with the RBD, whilst the second cluster interacts with the NTD and consists of six clonally related antibodies [27]. In the present study, one RBD cluster antibody (TRES6) and one NTD cluster antibody (TRES328) were used for the selection of viral escape mutations in vitro. Previously, we concluded that the NTD binding antibody (TRES328) interacts with the N5 loop of the NTD, whilst TRES6 interacts with the RBD, most likely at position T478. The presumed binding sites were identified through modelling but not confirmed experimentally [27]. Therefore, previously identified mutations within the RBD and NTD and additional viral escape mutations selected in the present study were cloned into spike expression plasmids. These plasmids were subsequently used for pseudotyping lentiviral vector particles used in neutralization assays and flow cytometry-based binding studies. This revealed that mutations at position S477 or T478 indeed affect binding of the RBD cluster antibodies. Surprisingly, neutralization mediated by NTD cluster antibodies was also affected by mutations within the RBD, whilst binding seemed unaffected. Variants resistant to the NTD cluster antibody, however, only displayed mutations within the NTD. RBD cluster antibodies remained highly effective against these variants. Since the antibodies bind to different epitopes and it was suggested previously that a combination therapy with an RBD and an NTD-binding antibody might be beneficial, the emergence of double-escape mutants was also analyzed [28].

## 2. Results

### 2.1. Generation and Identification of Escape Mutants

The authors observed previously that SARS-CoV-2 escape variants can arise against TRES6 and TRES328 antibodies, carrying either a T478K or a ΔL241-Y248;F mutation. To explore the role of these mutations and to identify additional SARS-CoV-2 epitopes that affect neutralization by TRES6 and TRES328, additional escape variants were selected. To this end, Vero-E6 cells were infected with early SARS-CoV-2 virus isolates (hCoV-19/Germany/ER1/2020 CoV-ER1 (GISAID: EPI_ISL_610249) or MUC-IMB-1 (GISAID EPI ISL 406862 Germany/BavPat1/2020)) in the presence of the 90% inhibitory concentration of TRES6 or TRES328, respectively. For the selection of a SARS-CoV-2 mutant resistant to both antibodies, each of the antibodies was used at half the 90% inhibitory concentrations. TRE6 and TRES328 antibodies were chosen from the two clusters as they display the lowest IC50 in the fully humanized form [27]. Cell culture supernatants were passaged on uninfected Vero-E6 cells in the presence of increasing TRES6 or TRES328 antibody concentrations for a total of five passages. Three independent viral isolates resistant to neutralization by TRES6 could be obtained (Figure 1A). In all three cases, the isolate was still neutralized by the NTD-binding antibody TRES328, reaching IC50s similar to the ones detected against the wild type virus [27]. Similarly escape variants generated against TRES328 were still neutralized by TRES6 (Figure 1B). The double escape variant, however, was no longer neutralized by either of the antibodies (Figure 1C). For the selection of the double escape mutant, it was necessary to add double the volume of the cell culture supernatant of the previous passage in order to maintain viral loads sufficiently high for continuous passaging (Figure 1D). Nevertheless, it was possible in all cases to obtain viral isolates for all antibodies and the antibody cocktail. In addition to the cultivation of the virus under selective antibody pressure, the virus was cultivated for five passages on cells without an antibody. This did not affect antibody neutralization (Figure 1C).

The viral escape variants were sequenced and mutations occurring with a frequency above 50% were considered to be emerging mutations. This way it was determined that three independently derived TRES6 escape variants carry either the S477N or a T478K or T478I mutation within the RBD (Figure 1E). Notably, all three escape variants to the RBD antibody carried a mutation within the NTD and a mutation within S1 in proximity to the cleavage site. The T478K isolate is additionally mutated at the position I68R and R682W, the T478I isolate at N74K and ΔQ675-N679, and the S477N variant carries a G261R and R682W mutation (Figure 1E). 

The TRES328 escape variants displayed mutations within the NTD. The first escape variant carries a deletion in the NTD ranging from L241 to Y248, with the entire site being substituted by an F. An additional deletion could be identified in this escape variant at position N679-A688. The other TRES328 escaping variant identified carries two mutations within the NTD, one deletion at position Y145 and a H245R mutation. Interestingly, the H245R mutation was also detected in the virus variant that was passaged for five rounds without an antibody, indicating again that this mutation is most likely linked to cell culture adaptations. The deletion at position at Y145, however, has also been shown for circulating VOC like the Alpha and Omicron variant [22,23,26]. The second variant resistant to TRES328 carried an additional deletion at the positions Q675-N679 (Figure 1F). Interestingly, the same deletion, in proximity to the fusion site, was also identified in the double escape variant (Figure 1G). This escape variant also carries a S477N mutation within the RBD and a D215G mutation within the NTD. The S477N mutation has also been detected before in the TRES6 escape mutation (Figure 1E), whilst the D215G mutation has been observed in the Beta variant of concern [24].

### 2.2. Functional Characterization of Mutations in Escape Variants of the RBD Cluster Antibody

To confirm the role of the different mutations observed in the escape variants, the mutations in the RBD identified through sequencing were cloned into an S expression plasmid to generate lentiviral pseudotypes for neutralization assays. Neutralization was assessed against the clonally related RBD cluster antibodies, TRES6, TRES224, and TRES567, and the NTD cluster antibodies, TRES49, TRES219, TRES328, TRES618, TRES1209, and TRES1293. As assumed, the T478I mutation leads to a complete abrogation of neutralization by RBD cluster antibodies (Figure 2A). The neutralization by NTD cluster antibodies, surprisingly, was also affected by the T478I mutation within the RBD, with none of the antibodies reaching neutralization values above 50% (Figure 2B). Antibody binding, however, is only impaired for the RBD cluster antibodies whilst NTD cluster antibodies still bind efficiently to the mutated spike (Figure 2C). Since the RBD escape variant with the T478I mutation also carried a N74K mutation, pseudotype neutralization assays were also performed with the N74K and T478I double mutant. While the double mutant was fully resistant to the RBD cluster antibodies, low level neutralization could be observed for some of the NTD cluster antibodies including TRES328 (Figure 2D,E). A similar observation was made for the T478K. Here again, neutralization is reduced, both for RBD and NTD cluster antibodies by the RBD mutation, whilst binding seems unaffected for the NTD cluster antibodies (Figure 2G,H,J). When the mutation detected within the NTD, the I68R mutation, is introduced in addition to the T478K mutation, the TRES328 antibody is able to neutralize at an IC50 of 2388 ng/mL (Figure 2J,K). Interestingly, even though the introduction of the N74K or I68R mutation improved neutralization by NTD cluster antibodies, full neutralization was never reached (Figure 2E,K). Binding of the antibodies is again only affected for RBD cluster antibodies whilst NTD cluster antibodies still bind efficiently to cells expressing the mutated S protein (Figure 2L). The elevated neutralization by cluster 1 antibodies observed in the graphs in Figure 2D,J seems to be of little functional relevance as none of the antibodies reach neutralizing levels of above 50% even at concentrations of 5 µg/mL. Introduction of the third identified RBD mutation, S477N, also leads to an abrogation of RBD cluster antibody binding, underlining that not only T478 but also S477 is crucial for the binding of RBD cluster antibodies and neutralization. NTD cluster antibodies did not neutralize the S477N-carrying pseudotyped particles (Figure 2M,N). The additional introduction of the G261R mutation into the NTD did, in contrast to the two mutations introduced previously, not improve neutralization (Figure 2P,Q). The binding pattern to S was, as initially expected, with the RBD cluster antibodies not able to bind, whilst the NTD cluster antibodies are (Figure 2O,R). The binding of the non-neutralizing antibody TRES1 is not affected by any of the mutations, consistent with its assumed binding site in the S2 domain of the S protein.

### 2.3. Functional Characterization of Escape Variants of the NTD Cluster Antibody

The two mutations identified in the TRES328 escape variants, ΔL241-Y248;F and ΔY145 were also introduced into a SARS-CoV-2 expression plasmid. This way the neutralization and binding capacity of antibodies towards SARS-CoV-2 S ΔL241-Y248;F or ΔY145 could be determined. As expected, lentiviral particles pseudotyped with either of the spikes carrying the NTD mutation are no longer neutralized or bound by any of the NTD cluster antibodies (Figure 3B,C,E,F). Thus, the shortening of the N5 loop by deletion of the amino acids L241-Y248 impairs binding and neutralization [27]. The deletion of position Y145, which has been observed frequently in viral variants that escape from NTD antibody-mediated neutralization, also blocks neutralization by TRES328 most likely due to an altered structure of the S protein by shortening of the NTD loop Y144-M153 [26,29]. Although the deletion of Y145 shortens the loop formed by the amino acids Y144-M153 by only one amino acid, it seems to be sufficient to change the interaction pattern of this loop to the TRES328 antibody. Besides Y145, this loop contains the two lysine residues K147 and K150, each of which has a strong electrostatic interaction with a glutamate residue of the antibody (Figure 3G). These lysine residues are shifted due to the deletion of position Y145 relative to the antibody, probably resulting in the weakening or loss of these strong atomic interactions. RBD cluster antibodies are still capable of neutralization (Figure 3A,D) with IC50s that are very similar to the IC50s determined against the wild type (D614G) isolate. Thus, these mutations within the NTD do not affect binding and neutralization by RBD cluster antibodies. Introduction of the H245R mutation into the NTD did neither affect neutralization and binding by RBD nor NTD cluster antibodies (data not shown). Again, binding of TRES1 is not altered by any of the mutations.

### 2.4. Functional Characterization of Mutations from an Escape Variant to RBD and NTD Cluster Antibodies

A variant resistant to both antibodies was more difficult to select than variants resistant to either of the antibodies alone. As indicated by the higher C_T_ values in the cell culture, supernatants during passaging virus replication were suppressed more efficiently and high inoculation volumes were used during passaging (Figure 1G). Nevertheless, we were able to obtain a double escape variant and identified two mutations that could be responsible for the viral escape (Figure 1C,G). One mutation is located within the NTD (D215G) and one within the RBD (S477N). When both mutations were introduced into an S expression vector, neither RBD nor NTD cluster antibodies were able to neutralize lentiviral particles pseudotyped with the mutated spike (Figure 4A,B). The mutations in the RBD and NTD not only affect neutralization, but also binding, which is impaired for all antibodies except the S2-binding antibody, TRES1 (Figure 4C). Although the D215G mutation is not located within the TRES328 interface, a mutation at this position to glycine can potentially induce a conformational change within the NTD, as stabilizing polar interactions with other amino acid side chains, such as a salt bridge with R214 (Figure 4D; PDB: 7C2L) [30] or a hydrogen bond with Y266 (PDB: 6VXX [31]), can no longer be formed.

### 2.5. Neutralization of VOC

In addition to the assessment of single-point mutations identified in the escape mutants, we analyzed the neutralization and binding capacity of the RBD and NTD cluster antibodies towards the spike proteins of variants of concern and interest and the D614G wild type virus that circulated early on during the pandemic. The latter is neutralized by antibodies of both clusters (Figure 5A,B). Again, none of the NTD cluster antibodies reached full neutralization, which is consistent with data obtained in whole-virus neutralization assays [27]. Binding to D614G is similar for all antibodies (Figure 5C). In accordance with the data obtained by analysis of the virus escape variants, RBD cluster antibodies only lose their neutralization capacity against the Delta variant, whilst S proteins of Gamma and Kappa variants are still neutralized and bound (Figure 5C,D,F,G,I,J,L). Neutralization by NTD cluster antibodies is impaired by all three variants tested (Gamma, Kappa and Delta). The partial loss of neutralization by NTD cluster antibodies, already noted for the TRES6 escape variants, was also observed against the Gamma VOC (Figure 5E). Here again, only TRES328 is able to neutralize the Gamma variant by at least 50%, leading to an IC50 of 527 ng/mL (Figure 5E). For the Kappa and Delta variant, neutralization by NTD cluster antibodies is completely abolished (Figure 5H,K). In contrast to the neutralization, the binding capacity of the NTD cluster antibodies to S of the Gamma or Kappa VOC is not reduced (Figure 5F,I), whilst binding to the Delta variant is impaired (Figure 5L).

## 3. Discussion

The functional characterization of antibody escape mutants can provide important information on the molecular mechanisms of virus neutralization and for selection and combination of antibodies for future clinical use. The results from the NTD cluster antibodies analyzed in the present study are consistent with results from other NTD interacting antibodies such as the 4A8 antibody [27,30]. These antibodies interact directly with the NTD and are able to prevent SARS-CoV-2 infection without blocking the RBD-ACE2 interaction. The neutralization mediated by this class of antibodies is most likely based on a steric hinderance of the viral entry process. This is supported by experimental and computational studies that describe the involvement of the NTD in the regulation of the conformation of the RBD. The exact mechanism, however, remains elusive to this day [19]. Nevertheless, it could be observed previously that the amino acid sequence of the RBD has an effect on the conformation of the NTD. It was shown for example that the introduction of the D614G mutation not only led to an altered RBD but also to a movement of the NTD [32]. Li et al. therefore argue that the NTD function is partially dependent on the conformation of the RBD and vice versa [20]. This argument is further underlined by the data presented here. We observed that the NTD cluster antibodies are no longer able to neutralize lentiviral particles pseudotyped with S477N, T478I, or T478K mutations in the RBD, although the NTD amino acid sequence remained unaltered and antibody binding could still be detected. The loss of neutralization is most likely due to an altered orientation of the NTD relative to the axis of the spike protein conveyed by the RBD mutation. NTD antibody binding may occur at a different angle of approach that no longer leads to steric hinderance of the viral entry process. Interestingly, the effect was partially reversed by the introduction of the NTD mutations, detected in the original viral escape variant, possibly partially reenabling the steric hinderance of the viral entry process. The mutations close to the S1 S2 fusion site were not assessed since they are located outside of the presumed binding site. Furthermore, mutations at the same or positions in close proximity were also observed in the control isolate passaged five times on Vero-E6 cells without antibody selection (data not shown), indicating that the mutations could be caused by passaging.

An altered and flexible orientation of the NTD described previously could also explain the incomplete neutralization of virus variants and lentiviral pseudotypes observed for the NTD cluster antibodies [27,32]. It has been hypothesized that the NTD predominantly interacts with the RBD as a wedge, thereby ensuring that the RBD remains in the upward conformation and is accessible for ACE2 binding [20,31]. Li et al., however, argue that the wedge can detach from the RBD, therefore enabling the free transition between the up and down conformation of the RBD. Both of these variants exist within one virus population [20]. This could imply that the NTD cluster antibodies are only able to neutralize SARS-CoV-2 when the NTD is in the wedged conformation that interacts with the RBD, as its binding is otherwise no longer able to sterically hinder the viral entry process. This steric hinderance also seems to be lost for the Kappa and Gamma virus variants as the S proteins of the two virus variants do not carry mutations within the binding site of the NTD cluster antibodies and are still bound by the antibodies, but no longer neutralized.

Another aspect that needs to be considered when looking at mutations within the SARS-CoV-2 S protein is the altered glycosylation patterns mediated by the mutation. Here the N74 position, also identified in an escape variant in this work, has been described early on during the pandemic as a putative glycosylation site that when mutated to a K leads to an ablated glycosylation. This altered glycosylation leads to differences in infectivity in comparison to the wild-type virus variant and may contribute to a modified spike glycan shield and therefore an altered recognition by antibodies [20,33,34]. The glycosylation of the NTD, however, does not only affect antibody recognition but also modulates the conformational dynamics and activity of the RBD [35]. This could also have an impact on antibody interaction due to a differently exposed NTD and RBD.

In addition, the results further suggest that a combinational therapy, comprising an NTD and an RBD-binding antibody can indeed improve neutralization breadth and more importantly heighten the barrier for escape variants as the emergence of those was only possible with a higher amount of input virus and a longer cultivation period. Despite this, our observation that only two point mutations can lead to a virus variant highly resistant to both antibodies questions whether the genetic barrier of this particular combination of antibodies is high enough. Clearly, the NTD and RBD domains substantially influence each other and this interaction should be taken into consideration when designing neutralizing antibody therapies. The design and usage of neutralizing antibodies that interact with NTD residues that are conserved among different viral isolates and different corona viruses could be beneficial. It was shown for example that the residues 159–170 and 230–233 are conserved among different SARS-CoV-2 isolates and corona viruses, hinting toward the fact that these sites are involved in the RBD–NTD interaction [20]. Concordantly, it is also of advantage to use RBD-binding antibodies that interact with conserved sites, like the CR3022 antibody that interacts with an epitope conserved between SARS-CoV-2 and SARS-CoV, which is located within the RBD [36].

Since antibody escape is a major selective pressure for future SARS-CoV-2 variants, having a broad panel of well-characterized neutralizing antibodies seems to be an important preparatory measure against future developments in the pandemic [37,38].

## 4. Materials and Methods

### 4.1. Antibodies Directed against SARS-CoV-2 Spike

Antibodies with murine Fc-portions recognizing the RBD (TRES6, TRES224, TRES567), NTD (TRES49, TRES219, TRES328, TRES618, TRES1209, TRES1293) or S2 (TRES1) epitopes of SARS-CoV-2 Spike were generated, purified from hybridoma supernatants, and checked for purity by Western Blot and Coomassie Gel as described previously [27,39,40].

### 4.2. Virus Propagation

SARS-CoV-2 virus propagation was performed as described before [27,41]. Briefly, virus isolates from SARS-CoV-2-infected patients hCoV-19/Germany/ER1/2020 CoV-ER1 (GISAID: EPI_ISL_610249) or MUC-IMB-1 (GISAID EPI ISL 406862 Germany/BavPat1/2020) were passaged twice on Vero-E6 cells kept in OptiPRO^TM^ (Thermo Fisher Scientific, Waltham, MA, USA) [27,42,43,44]. After sterile filtering the supernatant, the infectious titers were determined. To this end, 2 × 10^4^ Vero-E6 cells per well of a 96-well plate (Greiner, Kremsmünster, Austria) seeded on the previous day were infected with limiting dilutions of the virus. Three days after infection, the supernatant was removed, the cells washed with PBS, and fixed with 4% PFA (Morphisto, Frankfurt am Main, Germany) for 20 min. Next, the cells were permeabilized with 0.5% Triton X (Carl Roth, Karlsruhe, Germany) in PBS and blocked for 1 h with 5% skimmed milk in PBS. Afterwards, the cells were stained with a human anti-spike primary antibody mix or purified sera from a convalescent patient [27] diluted in 2% skimmed milk in PBS for 1h. After three washing steps, a secondary FITC goat-anti-human IgG antibody (#109-096-088, Jackson ImmunoResearch, West Grove, PA, USA) diluted in 2% skimmed milk in PBS was applied for 1 h. Subsequently, the cells were washed and the fluorescent signal assessed with a CTL-ELISPOT reader (Immunospot; CTL Europe GmbH, Bonn, Germany) and analyzed with ImmunoSpot^®^ fluoro-X™ suite (Cellular Technology Limited, Cleveland, OH, USA). The TCID50s were calculated as described previously [45].

### 4.3. Generation of Escape Mutants

For the generation of TRES6, TRES328, or TRES6 and TRES328, escape mutants 2 × 10^6^ Vero-E6 cells were seeded into T75 flasks (Greiner, Kremsmünster, Austria) on the day prior to the infection. For the infection, TRES6, TRES328, or a mix of equal parts TRES6 and TRES328 or no antibody were incubated with hCoV-19/Germany/ER1/2020 or MUC-IMB-1 at a TCID50 of 1 × 10^6^ in OptiPRO^TM^ for 1 h at 37 °C. MUC-IMB-1 was used for the first selection experiment with TRES6. All other mutants were generated by passaging hCoV-19/Germany/ER1/2020 CoV-ER1 in the presence of the antibodies. The respective IC90 of each antibody was chosen as the starting concentration and was then increased by a factor of two with every round of infection throughout the experiment. For the double-escape variant, the cells were incubated with half the IC90 of both antibodies. During the 1 h incubation period, the media was changed in the cell culture flasks to 14 mL fresh OptiPRO^TM^ (Thermo Fisher Scientific, Waltham, MA, USA) and after the incubation, 1.4 mL in the first round and 1 ml in the following rounds of the antibody-virus-mix were added to the cell culture. The cells were then checked daily for signs of infection. When cytopathic effects were visible, the supernatant was harvested, centrifuged for 5 min at 1200 rpm to remove cell debris, and then filtered through a 0.45 µm filter. The C_T_ value of the supernatant was determined by real time PCR and 100 µL supernatant (200 µL supernatant for the double-escape variant) was used for subsequent rounds of infection [46]. After five rounds of infection, the virus was sequenced and mutations identified as described earlier [27].

### 4.4. Neutralization Assays

For the assessment of viral neutralization 2 × 10^4^ Vero-E6 cells were seeded into 96-well plates 16–20 h prior to the experiment. On the day of the experiment, the antibodies of interest were serially diluted in OptiPRO^TM^ and incubated with the virus (1.8 × 10^5^ infectious units per well) for 1 h. Afterwards, the antibody virus mix was added onto the cells, and after 1 h, the medium was changed to fresh cell culture medium and the cells were incubated for another 20–24 h. Subsequently, the plates were washed and stained as described in Section 4.2 [27]. The IC50s were calculated with Prism 6 GraphPad (San Diego, CA, USA).

### 4.5. Generation of Spike Point Mutations

For the identification of spike mutants resistant to antibody neutralization, the mutations found in the escape virus variants were introduced as single or double point mutations into a SARS-CoV-2 spike expression plasmid. This plasmid encodes for the wild-type spike protein carrying a D614G mutation in the RBD [13]. Additionally, plasmids encoding for the Delta, Kappa, and Gamma spike variants were used [14,47]. For the assessment of antibody binding plasmids encoding a C-terminal HA tag were employed while for lentiviral pseudotype assays plasmids with a C-terminal 18-amino-acid-long truncation were used. Point mutations were introduced by site-directed mutagenesis PCR and oligonucleotide sequences for the introduction of point mutations. Plasmid sequences are available upon request. Sequences were checked with a commercially available sequencing service (EZ-seq, Macrogen, Amsterdam, The Netherlands).

### 4.6. Antibody Binding Test

For the analysis of antibody binding to mutated and wild-type SARS-CoV-2 Spike, HEK293T cells (ECACC 12022001) were transfected with plasmids encoding the spike protein carrying the respective mutation, an HA tag, and a BFP reporter plasmid, as described previously [27,48]. For this, 1 × 10^7^ cells were seeded in T75 flasks (Greiner, Kremsmünster, Austria) on the day prior to the transfection in 15 mL D10 (DMEM (Thermo Fisher Scientific, Waltham, MA, USA) supplemented with 10% Fetal Calf Serum (Capricorn, Ebsdorfergrund, Germany), 5 mL Glutamax (Thermo Fisher Scientific, Waltham, MA, USA), and 100 units/mL penicillin/streptomycin (Sigma-Aldrich, St. Louis, MO, USA)). On the day of transfection, the media in the flasks was changed to 10 mL DMEM without additives and the transfection mix was prepared by mixing 30 µg of a plasmid encoding for the spike of interest carrying a HA tag, 10 µg of a BFP-encoding plasmid, and 90 µL of polyethyleneimine (Sigma-Aldrich, St. Louis, MO, USA) in 2.5 mL DMEM without additives. Subsequently, the mixture was incubated for 15 min at RT and then added onto the cells. After 6–8 h of incubation, the supernatant was discarded and replaced by 15 mL fresh D10. After 48 h, the cells were harvested and 500,000 cells were seeded per well of a 96-well U-bottom plate. Thereafter, the cells were incubated with 100 ng/mL of the antibody of interest diluted in 50 µL of FACS buffer (0.5% Bovine Serum Albumin in PBS) for 30 min. After two washing steps, they were stained with an APC goat anti-mouse IgG antibody (#1030-31, Southern Biotech, Birmingham, AL, USA). Following two washing steps, the cells were fixed for 20 min with 2% PFA in PBS and then permeabilized for 10 min with 0.5% Saponin in FACS Buffer. Next, the cells were stained intracellularly for 30 min with an anti-HA FITC antibody diluted in FACS buffer (#NB600-363AF488, Novus Biologicals, Littleton, CO, USA) to assess the C-terminal HA tag expression. Finally, the cells were acquired on an FACS Attune Nxt (Thermo Fisher Scientific, Waltham, MA, USA) and analyzed with the software Flow Logic^TM^ (Inivai, Mentone, Australia). The binding indices were determined by the following formula: Binding index = (% TRES positive cells × MFI of TRES positive cells)/(% HA-positive cells × MFI of HA positive cells) × 100.

### 4.7. Lentiviral Pseudotype Assays

Lentiviral pseudotype neutralization was assessed as described previously [41,49,50,51]. The particles were pseudotyped with spike protein by transfecting the respective expression vectors containing the point mutations identified and generated in this publication. The IC50s were calculated with Prism 6 GraphPad (San Diego, CA, USA).

### 4.8. Structural Modeling

The structural modeling of TRES328 in complex with the NTD was performed as described previously [27]. Due to the high sequence identity between the antibody 4A8 and TRES328 (85% and 94% for the heavy and light chain, respectively), the crystal structure of the spike protein bound to antibody 4A8 (PDB: 7C2L [30]) was used for the structural modeling of TRES328 in complex with the wild type NTD. The structural modeling was done with Modeller 9.23 [52], and protein structure visualizations were made with UCSF Chimera 1.16 [53].

## Figures and Tables

**Figure 1 ijms-23-08177-f001:**
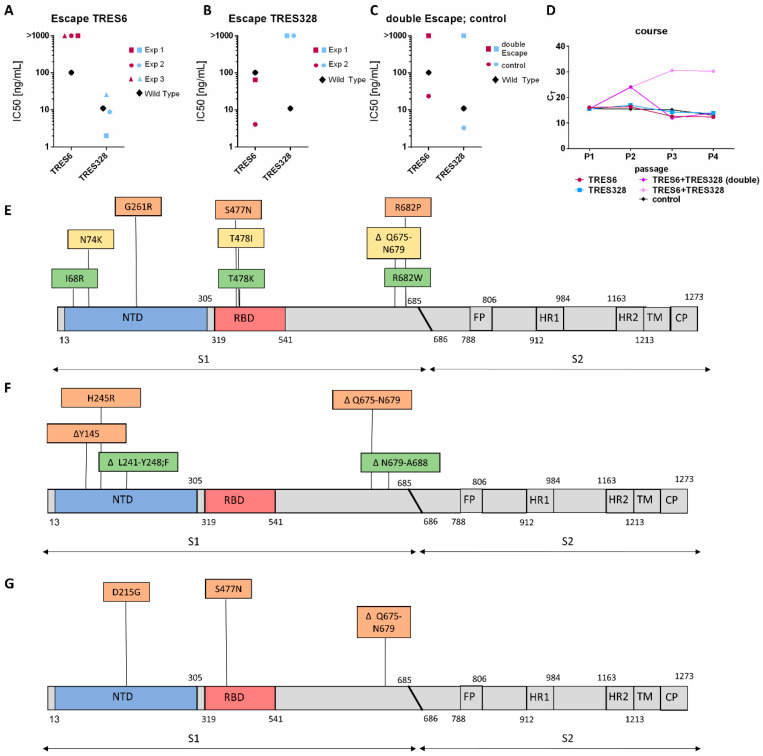
Neutralization of escape variants and identification of mutations. (**A**) IC50s of the antibodies indicated for the TRES 6 escape variants selected in three independent experiments (Exp) and the wild type virus. IC50s were taken from [27]; (**B**) IC50s of the antibodies indicated for TRES328 escape variants selected in two independent experiments and the wild type virus; (**C**) IC50s of the antibodies indicated for an escape mutant selected in the presence of TRES6 and TRES328 and without antibody and the wild type viral isolate; (**D**) viral load (C_T_) in cell culture supernatants of the indicated passages and antibodies used for selection. Double the inoculation volume was needed to maintain sufficient viral load levels during passaging in the presence of TRES6 and TRES328. (**E**–**G**) Mutations identified through amplicon sequencing of the TRES6 (**E**), TRES328 (**F**), and TRES6+TRES328 (**G**) viral escape variants. Mutations identified in the first isolate are displayed in yellow, in the second isolate in green, and the third isolate in orange. Map not drawn to scale. FP: fusion peptide; HR: heptad repeat; TM: transmembrane domain; CP: cytoplasmic domain.

**Figure 2 ijms-23-08177-f002:**
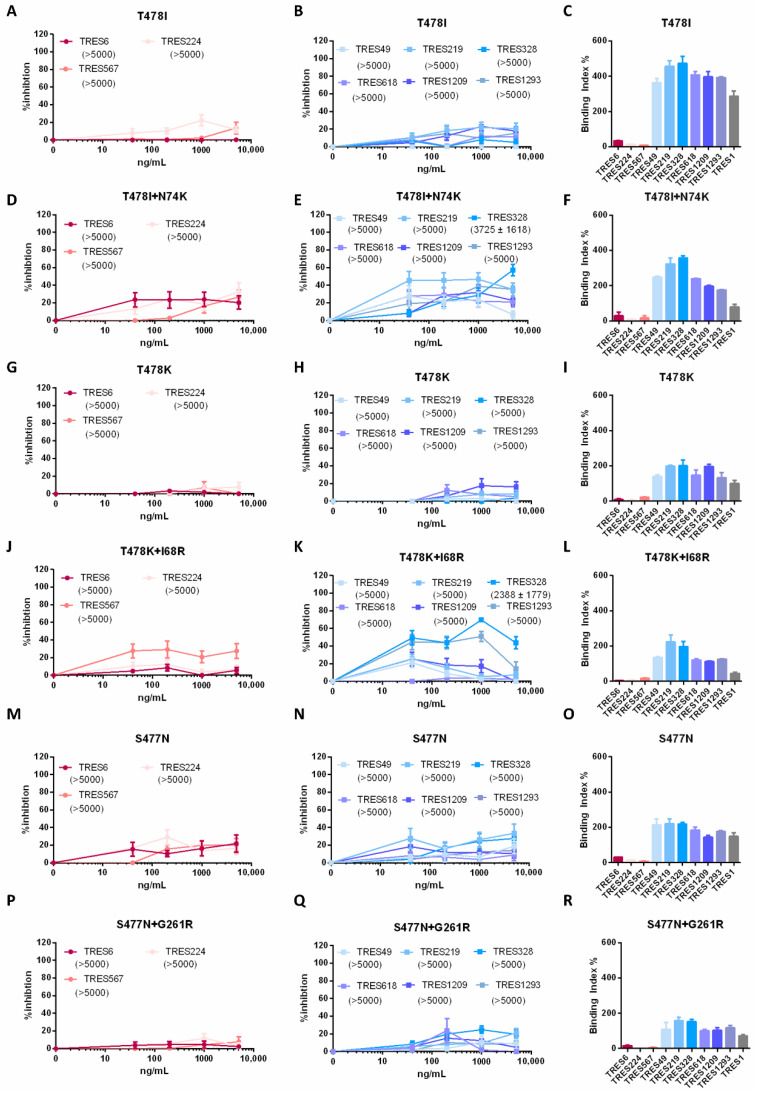
Characterization of S of TRES6 escape variants. (**A**,**B**,**D**,**E**,**G**,**H**,**J**,**K**,**M**,**N**,**P**,**Q**) Percent inhibition of lentiviral vector particles pseudotyped with SARS-CoV-2 S carrying the indicated RBD and/or NTD mutations. IC50s of the indicated RBD (red) and NTD (blue) cluster antibodies are displayed in ng/mL in the brackets. (**C**,**F**,**I**,**L**,**O**,**R**) Binding of the RBD (red) and NTD (blue) cluster antibodies and an S2-binding antibody (grey) to S with the indicated mutations. The binding capacity of cluster 1 and cluster 2 antibodies to HEK-293T cells expressing SARS-CoV-2 S proteins carrying a T478I (**C**), T478I+N74K (**F**), and T478K (**I**); or T478K+I68R (**L**), S477N (**O**), and S477N+G261R (**R**). Neutralization and binding of the control D614G variant is given in Figure 5A–C.

**Figure 3 ijms-23-08177-f003:**
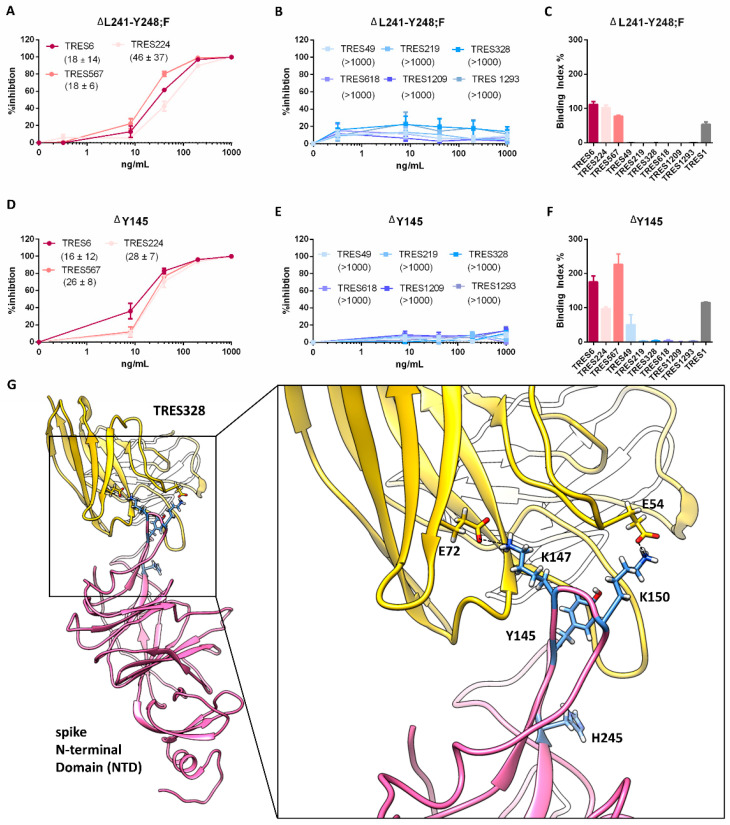
Characterization of S of TRES328 escape variants. (**A**,**B**,**D**,**E**) Percent inhibition of lentiviral vector particles pseudotyped with SARS-CoV-2 S carrying the indicated RBD and/or NTD mutations. IC50s of the indicated RBD (red) and NTD (blue) cluster antibodies are shown in brackets in ng/mL. (**C**,**F**) Binding of the RBD (red) and NTD (blue) cluster antibodies and a S2 binding antibody (grey) to S with the indicated mutations. Neutralization and binding of the D614G wild-type spike protein is given in Figure 5A–C. (**G**) Structural modeling of TRES328 (yellow) in complex with the NTD of wild-type spike protein (pink). NTD residues are represented as blue sticks and TRES328 residues as yellow sticks in order to highlight the position of Y145 and H245 or to illustrate the stabilizing electrostatic interactions (black dashed lines) between K147 and E72 of TRES328 and K150 and E54 of TRES328, respectively.

**Figure 4 ijms-23-08177-f004:**
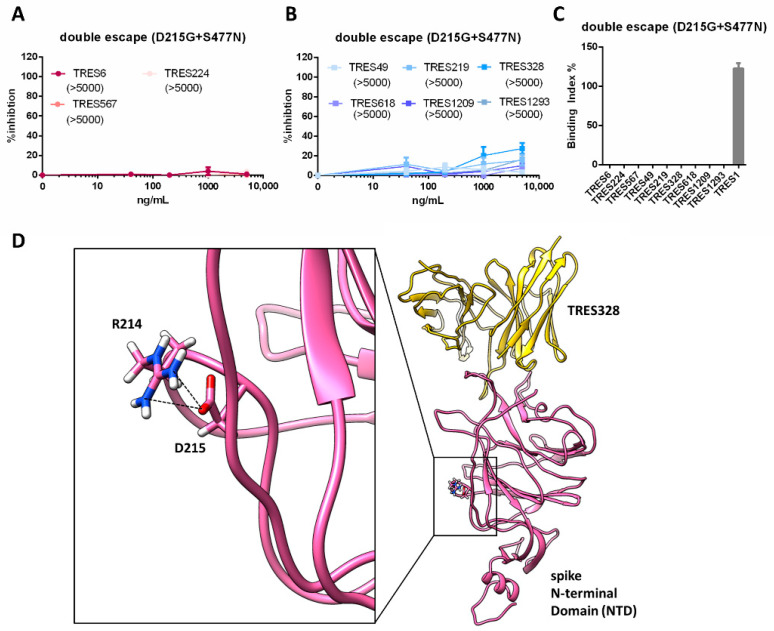
Characterization of S of the TRES6+TRES328 double escape variant. (**A**,**B**) Percent inhibition of lentiviral vector particles pseudotyped with SARS-CoV-2 S carrying the D215G and S477N mutations. IC50s of RBD (red) and NTD (blue) cluster antibodies are shown in ng/mL in the brackets. Neutralization and binding of the D614G wild-type spike protein is given in Figure 5A–C. (**C**) Binding of the RBD (red) and NTD (blue) cluster antibodies and an S2 binding antibody (grey) to S with the D215G and S477N mutations. (**D**) Structural modeling of TRES328 (yellow) in complex with the NTD of wild-type spike protein (pink). The NTD residues R214 and D215 are shown as sticks to illustrate that D215 is not located within the TRES328 interface and to highlight the close proximity of the R214 and D215 side chains, which form in some structures a salt bridge as strong electrostatic and stabilizing interaction (black dashed lines). The mutation to glycine (D215G) destroys this salt bridge interaction with R214, probably resulting in an increased flexibility of the arginine side chain, which could thereby possibly lead to a conformational change of the NTD.

**Figure 5 ijms-23-08177-f005:**
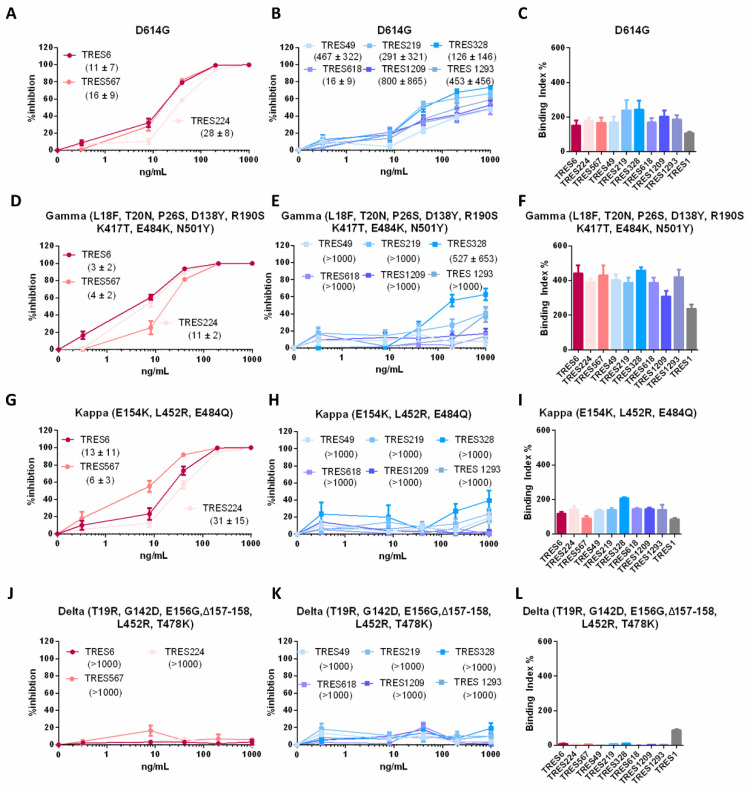
Characterization of SARS-CoV-2 S of the indicated VOCs. (**A**,**B**,**D**,**E**,**G**,**H**,**J**,**K**) Percent inhibition of lentiviral vector particles pseudotyped with S of the indicated VOCs. The relevant mutations within the NTD and RBD of the respective mutation are given in brackets. IC50s of the indicated RBD (red) and NTD (blue) cluster antibodies are shown in ng/mL in the brackets. (**C**,**F**,**I**,**L**) Binding of the RBD (red) and NTD (blue) cluster antibodies and an S2-binding antibody (grey) to S of the VOCs indicated.

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
