# Peer review of "Characterization of SARS-CoV-2 Escape Mutants to a Pair of Neutralizing Antibodies Targeting the RBD and the NTD"

_ijms, 2022, doi:10.3390/ijms23158177_

Round 1
Reviewer 1 Report
In this study, Antonia Sophia Peter et al. identified escape mutants of SARS-CoV-2 against two of their previously discovered antibodies. Mutations occurred at two RBD positions were able to confer resistance against both RBD and NTD cluster antibodies. Following are some comments.
Previously, the authors identified 3 RBD antibodies and 6 NTD antibodies. Are the CDRs of the antibodies from same cluster similar or completely different? And why TRES6 from RBD antibodies and TRES328 from NTD antibodies were selected for the generation of escape mutants?
Did the authors try to combine the mutations in escape variants of the RBD cluster antibody and NTD cluster antibody to generate new pseudoviruses, and test whether these pseudoviruses can escape both cluster antibodies.
Figure2 Introduction of the N74K or I68R mutation improved neutralization by NTD cluster antibodies, and there is also a slightly increased inhibition of the RBD cluster antibodies. The NTD mutation was detected along with the RBD mutation in the escape variants of the RBD cluster antibody, why pseudovirus with double mutation had a weaker escape ability than the pseudovirus with a single RBD mutation?
Figure5 The important RBD or NTD mutations of the Gamma, Kappa and Delta variant compared to the original SARS-CoV-2 could be labeled in the figure to give a better presentation.
Reviewer 2 Report
Comments:
1. line109, since the authors tested resistant mutations for all other TRES antibodies in following result, it might be necessary to figure out more clearly why TRES6 and TRES328 were selected for escap mutation assay
2. line117:
(1) why three exp for TRES6, but 2 for TRES328, and one for the cocktail? does that mean some isolates didn't generate escape mutants?
(2) need to point out in the manuscript which isolate was used in which independent experiment
3. line123-125, did the authors keep the Ab concentration consistent as before?
4. Figure 1, fig 1A, 1B, 1C, suggest to add the IC50 against WT virus into the fig for comparison
5. line171-173, but how the authors explain that in fig 1A, TRES328 wasn't impacted by RBD mutations (described in line119-120), given the mutation exists in these resistane isolates
6. line201, fig 2, all figs need control group (without mutations) for comparison. Same comment for fig 3/4/5
7. line245-246, what about N679-A688, Q675-N679? looks they are ignored in the manuscript
Round 2
Reviewer 2 Report
Thanks for the authors addressing all questions.